

# Nitro-polycyclic aromatic hydrocarbons - gas-particle partitioning, mass size distribution, and formation along transport in marine and continental background air

Gerhard Lammel [1,2*], Marie D. Mulder [1], Pourya Shahpoury [2], Petr Kukučka [1], Hana Lišková [1], Petra Přibylová [1], Roman Prokeš [1], Gerhard Wotawa [3]

[1] Masaryk University, Research Centre for Toxic Compounds in the Environment, Brno, Czech Republic

[2] Max Planck Institute for Chemistry, Multiphase Chemistry Department, Mainz, Germany

[3] Zentralanstalt fuer Meteorologie und Geodynamik, Wien, Austria

*lammel@recetox.muni.cz

**Abstract**

Nitro-polycyclic aromatic hydrocarbons (NPAH) are ubiquitous in polluted air but little is known about their abundance in background air. NPAHs were studied at one marine and one continental background site i.e., a coastal site in the southern Aegean Sea (summer 2012) and a site in the central Great Hungarian Plain (summer 2013), together with the parent compounds, PAHs. A Lagrangian particle dispersion model was used to track air mass history. Based on Lagrangian particle statistics, the urban influence on samples was quantified for the first time as a fractional dose to which the collected volume of air had been exposed to. At the remote marine site, the 3-4 ring nitro-PAH (sum of 11 targeted species, with 2-nitrofluoranthene (2NFLT) and 3-nitrophenanthrene being the most abundant) concentration was 23.7 pg m$^{-3}$ while the concentration of 4-ring PAHs (6 species) was 426 pg m$^{-3}$. Urban fractional doses in the range <0.002–5.4% were received. At the continental site, the $\Sigma_{11}$ 3-4rNPAH and $\Sigma_6$ 4rPAH were 58 and 663 pg m$^{-3}$, respectively, with 9-nitroanthracene and 2NFLT being highest concentrated amongst the targeted NPAHs. The NPAH levels observed in the marine background are the lowest ever reported and remarkably lower, by more than one order of magnitude, than one decade before. Day-night variation of NPAHs at the





continental site reflected shorter lifetime during the day, possibly because of photolysis of
some NPAHs. The yields of formation of 2NFLT and 2NPYR in marine air seem to be close
to the yields for OH-initiated photochemistry observed in laboratory experiments under high
$NO_x$ conditions. Good agreement is found for prediction of NPAH gas-particle partitioning
using a multi-phase poly-parameter linear free energy relationship. Sorption to soot is found
less significant for gas-particle partitioning of NPAHs than for PAHs.
The NPAH levels determined in the southeastern outflow of Europe confirm intercontinental
transport potential.

**Keywords:** long-range transport potential, semi-volatile organic compounds, PAH
photochemistry,

**1. Introduction**
PAHs may undergo chemical transformations in the gaseous and in the particulate phase
(Finlayson-Pitts and Pitts, 2000; Keyte et al., 2013). Nitro-PAHs (NPAHs), earlier predicted
based on smog-chamber experiments (Atkinson and Arey, 1994), and later observed in urban
and rural areas (Finlayson-Pitts and Pitts, 2000; Keyte et al., 2013), seem to be most
significant derivatives: Mutagenicity and toxicity of atmospheric aerosols in general is mostly
related to NPAHs (Grosjean et al., 1983; Garner et al., 1986; Finlayson-Pitts and Pitts, 2000;
Claxton et al., 2004; Hayakawa, 2016). A large part, more than one third of the mutagen
potential of ambient may be attributable to NPAHs (Schuetzle, 1983).
Secondary formation of NPAH from PAHs is thought to occur on short time scales (hours).
This has been observed for PAHs collected on filters (Ringuet et al., 2012a; Zimmermann et
al., 2013; Jaryasopit et al., 2014a, 2014b), and also in in urban plumes (Bamford and Baker,
2003; Arey et al., 1989; Reisen and Arey, 2005). Although many NPAHs are emitted from
road traffic, only few are abundant in this source type (Arey, 1998; Keyte et al., 2013 and
2016; Inomata et al., 2015; Alves et al., 2016). The occurrence of various isomers of
nitrofluoranthene (NFLT) and nitropyrene (NPYR) can be used to study PAH sources, PAH
chemical transformations and the role of the photo-oxidants hydroxyl radical (OH) and nitrate
radical ($NO_3$) (Ciccioli et al., 1996; Finlayson-Pitts and Pitts 2000). E.g., 3- and 2-
nitrofluoranthene (3-, 2NFLT) are indicative for primary and secondary sources, respectively.



These substances have been suggested as tracers for air pollution on the time scales of hours
to days (Ciccioli et al., 1996; Finlayson-Pitts and Pitts 2000; Keyte et al. 2013).
Like their precursors, NPAHs are SVOCs, partitioning between the phases of the atmospheric
aerosol. Similar to other SVOCs, the NPAHs' phase distribution was found to depend on
temperature (summer and winter campaigns in the Alps; Albinet et al., 2008) and results from
both absorptive as well as adsorptive contributions (Tomaz et al., 2016). NPAHs have
primarily been observed in polluted areas (e.g. Pitts et al., 1985; Ramdahl et al., 1986; Garner
et al., 1986; Albinet et al., 2006 and 2008a; Ringuet et al., 2012a and 2012b; Zimmermann et
al., 2012; Barrado et al., 2013; Li et al., 2016). However, NPAHs have hardly been studied in
the unpolluted environment: Few studies were conducted in the rural environment i.e., in
Germany (Ciccioli et al., 1996), in the French Alps (100-1000 pg m$^{-3}$ range for the sum of 10
NPAHs; Albinet et al., 2008) and in northern China (Li et al., 2016). Very few measurements
were performed in the remote atmospheric environment i.e., in the Mediterranean (Tsapakis
and Stephanou, 2007), in the Himalayas (single data; Ciccioli et al. 1996), and in the Arctic
(with so-called Arctic haze; Masclet et al. 1988; Halsall et al. 2001). With regard to the long-
range transport potential, the state of the knowledge is that at least some NPAHs are suspect
to go into intercontinental transport (Lafontaine et al., 2015) and might be ubiquitous in the
global atmosphere (Ciccioli et al.,1996).
However, a lack of NPAH data from remote atmospheric environments is obvious and little is
known about their long-range transport potential. The aim of this study was to study the long-
range transport potential of NPAHs by measurements at remote sites of Europe, addressing
the continental background and the outflow of the continent.

**2. Methodology**
**2.1 Sampling**
High-volume air sampling was conducted at a marine background site, Finokalia
(35.3°N/25.7°E, 250 m a.s.l.), in the context of a coordinated field experiment 2-13 July 2012
(Lammel et al., 2015) and at a continental background site in central Europe, K-puszta
(46º58'N/19º33'E, 125 m a.s.l.; Degrendele et al., 2016), 5-16 August 2013. The Finokalia site
is located on a cliff at the northern coast of Crete, some 70 km east of major significant
anthropogenic emissions (Iraklion, a city of 100000 inhabitants with airport and industries;
Mihalopoulos et al., 1997; Kouvarakis et al., 2000). The K-puszta site is located on a clearing,
characterised by uncultivated grassland, in a mostly coniferous forest in the Hungarian





(Pannonian) Great Plain, ca. 70 km and 270 km southeast of Budapest and Vienna,
respectively (≈2 mn inhabitants each). The background site character of both observatories
had been demonstrated (Borbély-Kiss et al., 1988; Kouvarakis et al., 2000; Vrekoussis et al.,
2005). Meteorological and trace gas measurements are covered by both observatories, which
are stations of the EMEP network (EMEP, 2015).
High volume air samples were collected using a HV-100P (Baghirra, Prague, Czech
Republic), equipped with a multi-stage cascade impactor (Andersen Instruments Inc.,
Fultonville, New York, USA, series 230, model 235) with five impactor stages, corresponding
to 10–7.2, 7.2–3, 3–1.5, 1.5–0.95 and 0.95–0.49 μm of aerodynamic particle size, D, (spaced
roughly equal $\Delta$logD), a backup filter collecting particles < 0.49 μm and, downstream, two
polyurethane foam plugs (PUFs, Molitan, Břeclav, Czech Republic, density 0.030 g cm$^{-3}$,
placed in a glass cartridge), together 10 cm high. Particles were sampled on slotted QFF
substrates (TE-230-QZ, Tisch Environmental Inc., Cleves, USA, 14.3 × 13.7 cm) and glass
fibre filters (Whatman, 20.3 × 25.4 cm). The filters had been cleaned prior to use by heating
(330°C). PUFs were cleaned (8 hour-extraction in acetone and 8 hours in dichloromethane
(DCM)), wrapped in two layers of aluminum foil, placed into zip-lock polyethylene bags and
kept in the freezer prior to deployment. The sampler was operated at constant flow rate of 68
m$^3$ h$^{-1}$. Day/night sampling (changing at sunset and sunrise) of gaseous samples (PUF) was
performed at both sites (V = 600-1000 m³), while at the marine site the impactor filter (QFF)
samples were collected over 24 h (5) or 48 h (3).
Particle number concentration was determined by an optical particle counter (Grimm model
107, Ainring, 31 channels between 0.25 and 32 mm of aerodynamic particle diameter, D).
Aerosol surface concentration, S (cm$^{-1}$), was derived as $S = \pi \Sigma_i N_i D_i^2$ assuming sphericity.
Hereby, true S will be underestimated, in particular if particles of irregular form were
abundant (e.g. Jaenicke, 1988). Comparisons with absolute methods (e.g. Pandis, et al. 1991)
suggest that the discrepancy may reach up to a factor of 2-3. The mass median diameter ($D_m$,
μm), was derived as $\log D_m = \Sigma_i m_i \log D_i / \Sigma_i m_i$ with $m_i$ denoting the mass in size class i, $D_i$
being the geometric mean diameter collected on stage i of the cascade impactor.

**2.2 Chemical analysis**
All air samples were extracted with DCM using an automatic warm Soxhlet extractor (Büchi
B-811, Switzerland). Deuterated PAHs (D8-naphthalene, D10-phenanthrene, D12-perylene;





Wellington Laboratories, Canada) were used as surrogate standards for both PAHs and
NPAHs. These were spiked on each PUF prior to extraction. The extract was split in two
parts, 1/9 for PAHs and Nitro-PAHs analysis, 9/10 for PBDEs, PCBs and OCPs. The PAHs
and Nitro-PAHs aliquot was a subject to open column chromatography clean-up. Glass
column (1 cm i.d.) was filled with 5 g activated silica (150°C for 12 h), sample was loaded
and eluted with 10 mL $n$-hexane, followed by 40 mL DCM. The cleaned sample was
evaporated under a stream of nitrogen in a TurboVap II apparatus (Biotage, Sweden),
transferred into a conical GC vial and spiked with recovery standard, terphenyl, the volume
was reduced to 100 µL.
GC-MS analysis of 4-ring PAHs (fluoranthene (FLT), pyrene (PYR), benzo(b)fluorene
(BBN), benzo(a)anthracene (BAA), triphenylene (TPH) and chrysene (CHR)) and 2-4 ring
NPAHs (1- and 2-nitronaphthalin (1-, 2NNAP), 3- and 5-nitroacenaphthene (3-, 5NACE), 2-
nitrofluorene (2NFLN), 9-nitroanthracen (9NANT), 3- and 9-nitrophenanthren (3-, 9NPHE),
2- and 3-nitrofluoranthene (2-, 3NFLT), 1- and 2-nitropyrene (1-, 2NPYR), 7-
nitrobenz(a)anthracene (7NBAA), 6-nitrochrysene (6NCHR) was performed using a gas
chromatograph atmospheric pressure chemical ionization tandem mass spectrometer (GC-
APCI-MS/MS) instrument, Agilent 7890A GC (Agilent, USA), equipped with a 60m ×
0.25mm × 0.25um DB-5MSUI column (Agilent, J&W, USA), coupled to Waters Xevo TQ-S
(Waters, UK). Injection was 1 µL splitless at 280°C, with He as carrier gas at constant flow
1.5 mL min$^{-1}$. The GC oven temperature program was as follows: 90°C (1 min), 40°C/min
to150°C, 5°C/min to 250°C (5 min) and 10°C/min to 320°C (5 min). APCI was used in
charge transfer conditions. The isomers 2- and 3NFLT were not separated by the GC method,
but co-eluted and are reported as sum.
Recovery of native analytes varied 72-102% for PAHs and deuterated PAHs, 70-110% for
NPAHs (details see supplementary material (SM), Table S1a). The results were not recovery
corrected. The mean of field blank values was subtracted from the sample values. Values
below the mean + 3 standard deviations of the field blank values were considered to be
<LOQ. 3 field blanks of the sampling media types (slotted and backup QFF, PUF) were
collected at each site. Field blank values (listed in SM, Table 1c) of some analytes were below
the instrument limit of quantification (ILOQ), which corresponded to 0.004-0.069 pg m$^{-3}$ for
NPAHs (except for 1NNAP for which it ranged 0.60-0.87 pg m$^{-3}$) and 0.010-0.126 pg m$^{-3}$ for
4-ring PAHs (except for FLT and PYR for which it ranged 0.17-0.59 pg m$^{-3}$) (Table S1b).

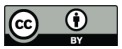



Higher LOQs were determined for some of the NPAHs and for all 4-ring PAHs in gaseous air
samples (PUFs), namely 0.006-0.009 ng (corresponding to 3.5-8.0 pg m$^{-3}$) for 3NACE and
2NPYR, 0.028-0.097 (corresponding to 16-86 pg m$^{-3}$) for 2NNAP, 2NFLT and 1NPYR, and
0.10-0.27 ng (corresponding to ≈60-240 pg m$^{-3}$) for 4-ring PAHs (except for FLT and PYR for
which it was 1.71 and 1.05 ng, respectively, corresponding to ≈600-1500 pg m$^{-3}$). In
particulate phase samples, where separate field blanks for the 2 different QFFs were
determined (on the impactor stages on one hand side and the backup filter on the other hand
side, Tbale S1c), higher LOQs were determined for some of the NPAHs and for all 4-ring
PAHs, namely 0.008-0.089 ng (corresponding to 4.6-79 pg m$^{-3}$) for 2NNAP, 2NFLT, 1NPYR
and 2NPYR, 0.26-0.31 ng (corresponding to 150-274 pg m$^{-3}$) for 9NANT, and 0.05-0.22 ng
(corresponding to ≈30-200 pg m$^{-3}$) for 4-ring PAHs (except for FLT and PYR for which it was
0.79 and 0.36 ng, respectively, corresponding to ≈200-700 pg m$^{-3}$).
The breakthrough in PUF samples was estimated (Pankow, 1989; ACD, 2015; Melymuk et
al., 2016), and as a consequence, 2-3 ring PAHs and 2-ring NPAHs results were excluded
from this study as their sampling may have been incomplete. We, therefore, report $\Sigma_6$4rPAH
and $\Sigma_{11}$3-4rNPAH.
Particulate matter mass (PM$_{10}$) was determined by gravimetry, and organic matter (OM) and
elemental carbon (EC) contents of PM by a thermal-optical method (Sunset Lab., USA;
EUSAAR protocol).

**2.3 Gas-particle partitioning**
Gas-particle partitioning was studied by applying a multiphase ppLFER model, which was
recently introduced (Shahpoury et al., 2016). In brief, partitioning of semivolatile compounds
in air can be described (Yamasaki et al., 1982), by

$K_p = c_{ip} / (c_{ig} \times c_{PM})$

where $K_p$ (m$^3$air (g PM)$^{-1}$) is the temperature dependent partitioning coefficient, $c_{PM}$ (g m$^{-3}$) is
the concentration of particulate matter in air, $c_{ip}$ and $c_{ig}$ are the analyte (i) concentrations (ng
m$^{-3}$) in the particulate and gas phase, respectively. $K_p$ can be predicted using models based on
single- and poly-parameter linear free energy relationships (spLFER, ppLFER). spLFER's
relate the partitioning coefficient to one physic-chemical property i.e., assume one process to





determine the sorption process, while ppLFER's in principle account for all types of
molecular interactions between solute and matrix (Goss and Schwarzenbach, 2001). The
observed particulate mass fractions were tested with both a spLFER and a ppLFER model.
The spLFER chosen is the widely used $K_{oa}$ model of Finizio et al., 1997 (results presented in
the SM, S2.3). The ppLFER is a multi-phase model recently presented (Shahpoury et al.,
2016) and applied for NPAHs (Tomaz et al., 2016). It is based on linear solvation energy
relationships (Abraham, 1993; Goss, 2005):

$\log K_p = eE + sS + aA + bB + lL + c$
$\log K_p = sS + aA + bB + vV + lL + c$

where capital letters E, S, A, B, L, and V are solute-specific Abraham solvation parameters for
excess molar refraction (describes interactions between π- and lone (n-) electron pairs),
polarizability/ dipolarity, solute H-bond acidity, solute H-bond basicity, logarithm of solute
hexadecane-air partitioning coefficient (unitless), and McGowan molar volume (cm$^3$
mol$^{-1}$)/100, respectively (Endo and Goss, 2014). The corresponding parameters e, s, a, b, l,
and v reflect matrix-specific solute-independent contribution to $K_p$. In lack of experimental
data, the solute descriptors for NPAHs were taken from M.H. Abraham (personal
communication). The multi-phase ppLFER considers adsorption onto soot, $(NH_4)_2SO_4$, and
$NH_4Cl$, and absorption into OM. OM is assumed to be constituted of two separate phases, low
to high molecular mass, both organic and water soluble OM, represented by a ppLFER
equation for dimethyl sulfoxide-air on one hand side, and high molecular mass OM,
represented by a ppLFER equation for polyurethane ether-air (Shahpoury et al., 2016).
A conventional single-parameter LFER ($K_{oa}$) model is applied, too.

**2.4 Air mass history analysis**
The HYSPLIT (Draxler and Rolph, 2003) and FLEXPART (Stohl et al., 1998, 2005) models
were used to identify air mass histories over 10 and 2 days, respectively. The possible
influence of polluted air on samples was quantified using a novel method of applying
Lagrangian particle statistics (FLEXPART, see SM, S2.2). To this end, for the entire sampling
period, one particle per second was released. The model output is generated at 0.062° (≈7
km), every 30 minutes and expressed as 'residence time' i.e., a measure of the time particles



resided in grid cells. ECMWF meteorological data (0.125°×0.125° resolution, hourly) were
used as input.

**3. Results and discussion**
The NPAH levels are distinctly lower at the marine than at the continental site, $\Sigma_{11\ 3\text{-}4r}$ NPAH =
22.5 and 58.5 pg m$^{-3}$, respectively (Table 1). The NPAHs showing the highest concentrations
were 2NFLT and 3NPHE at the marine (Fig. 1b) and 9NANT and 2NFLT at the continental
site (Fig. 1d, Table 2). The substance patterns at both sites are similar, though (R² = 0.76, P >
0.99, t-test). At the marine site, advection was northerly, with air masses originating (time
horizon 10 days) in eastern and central Europe and, towards the end of the campaign, in the
western Mediterranean. The site was placed into the southeastern outflow of Europe. NO$_x$
(0.2-0.6 ppbv), EC (0.2-0.8 µg m$^{-3}$) and PM$_{10}$ (18.3-39.3 µg m$^{-3}$) reflect background
conditions. Air mass history analysis suggests that the somewhat elevated concentration in the
first sample collected at the marine site (Fig. 1a) is related to long-range transport influenced
by passage over the urban areas of Izmir and Istanbul (urban fractional dose $D_u$ = 5.0%, in
contrast to the mean which was 1.6%; Fig. S3). Overall, urban fractional dose in the range
<0.002–5.4% was received at the marine site. Across all samples at the marine site, $D_u$ is
found to be significantly correlated with the pollutant sum concentrations $\Sigma_{6\ 4r}$ PAH and $\Sigma_{11\ 3\text{-}4r}$
NPAH (R² = 0.61 and 0.69, respectively, both P > 0.99).
From this data set, subsets of each two samples are formed, representing minimum (i.e.,
almost no influence from industrialised area 48 hours prior to arrival (herefoth called 'marine
background', urban fractional dose $D_u$ = 0.4%) and maximum observed influence (herefoth
called 'background with urban influence', $D_u$ = 3.1%; SM Table S2, Figure S3). The results
for these subsets are listed in Tables 1-3. Such classification was not deemed meaningful for
the samples collected at the continental site, as the relevant source distribution in central
Europe was too homogeneous during this episode. Advection was mostly from northwest and
partly from easterly directions, with air mass origin (time horizon of 10 days) mostly in central
Europe and, to a lesser extent in eastern Europe and the western Balkans. The NO$_2$ (1.2-2.6
ppbv), total carbon (3-6 µg m$^{-3}$) and PM$_{10}$ (10.7-46.3 µg m$^{-3}$) levels during the campaign
reflect continental background conditions.
The 4-ring PAH concentrations in samples from the continental site on the one hand, and in
background air with urban influence collected at the marine site (urban areas 300-500 km



away) on the other hand, are similar (Table 2). Also, the substance patterns are more similar
than when relating all samples at the marine site i.e., $R^2 = 0.88$ (P > 0.999, t-test) instead of $R^2$
= 0.76. At both sites, 4rPAHs were in fact influenced by secondary emissions, namely
throughout day and night from the soil (by average 16.3 and 9.3 pg m$^{-2}$ h$^{-1}$ for FLT and PYR,
respectively; Degrendele et al., 2016) or occasionally from surface seawater (during at least 1
day-time interval out of in total 3 of this data subset; Lammel et al., 2016). In the data set from
the continental site, we study day/night (D/N) effects (subsets listed in Tables 1-3, too): PAH
concentrations were ≈60% higher during the day than during the night, while NPAH
concentrations were by average ≈5% lower during the day. PAH concentrations were driven
by re-volatilisation from soil, determined by temperature variation (Degrendele et al., 2016).
For NPAHs (partly primary emitted) this indicates that the higher emissions during the day
(due to re-volatilisation and road traffic) were compensated by lower lifetime (because of
heterogeneous photolysis; Fan et al., 1996; Feilberg and Nielsen, 2000, 2001; García-Berríos
et al., 2017). The same could be reflected in different NPAH/PAH ratios (the potential NPAH
yields), which were 5.6% and 8.9% at the marine and continental sites, respectively. These
were influenced by similar substance patterns upon emission, similar irradiation (summer, no
or almost no clouds) and deposition velocities (θ in the range 0.05-0.20 for $\Sigma_{11}$ 3-4rNPAH and
$\Sigma_6$ 4rPAH, no precipitation), but different re-volatilisation fluxes and different characteristic
transport times elapsed. Distance to major urban source areas was 300- >1000 km at the
marine and 100-500 km at the continental site. The NPAH/PAH ratios being lower at the
more distant receptor site, the marine site, may suggest that photochemical degradation of
NPAHs along transport was on average faster than degradation of the precursors.

The NPAH levels observed in marine background air are the lowest ever reported.
Remarkably, the concentrations are much lower, by more than one order of magnitude, than
one decade before at the same site during the same season (Tsapakis and Stephanou, 2007).
The concentrations observed now are a factor of 4-10 lower than in a forest site in Amazonia
two decades before (which might have been influenced by biomass burning emissions), and
also a factor of 3 lower (for 2NPYR) than observed at an extremely remote site in the
Himalayas two decades before (Ciccioli et al., 1996; Table 3). The NPAH levels observed at
the marine site with influence of pollution and at the continental site are comparable, but also



at the lower end of the range spanned by previous observations at rural and remote sites
(Table 3).

**Gas-particle partitioning**
The time-weighted mean NPAH phase distributions ($\Sigma_{11}$3-4rNPAH) differ, corresponding to $\theta$
= 0.05 and 0.17 at the marine and continental sites, respectively, – despite similar
temperatures (Table 1). In contrast and despite of similar temperature ranges, the 4-ring
PAHs' ($\Sigma_6$4rPAH) particulate mass fraction was higher at the marine than at the continental
site ($\theta$ = 0.42 and 0.20, respectively). The NPAH mass size distribution had its maximum in
the <0.49 μm size range at both sites. The 4-ring PAHs mass size distribution had 2 maxima,
<0.49 μm and between 0.95 and 1.5 μm, at the marine site, but one at <0.49 μm at the
continental site (Table 1). This is probably related to the presence of aged aerosol at the
marine site vs. a larger contribution of fresh aerosols at the continental site. This is,
furthermore, supported by the analysis of air mass origins that shows significant influence of
urban areas for only few samples at the marine and for all samples at the continental site (SM
S2).
Both 4-ring PAHs and 3-4 ring NPAHs were more associated with PM in polluted air than in
clean air. This trend is weak for PAHs with $\theta$ = 0.02 for $\Sigma_6$4rPAH in marine background but
0.07 in background with urban influence (and $\theta$ = 0.09 and 0.25 for CHR; Table 2), but is
obviously strong for NPAHs, namely $\theta$ = 0.19 for 2NPYR in marine background but 0.70 in
background with urban influence, ≈0.93 in polluted continental air, and $\theta$ = 0.01 for $\Sigma_{11}$3-
4rNPAH in marine background but 0.21 in background with urban influence (Table 2). The
urban influenced air at the marine site is also reflected in a much higher OC (a factor of 3
higher than the all-campaign mean) and elevated EC, (less prominent, ≈50% above mean).
This confirms the understanding that gas-particle partitioning of both PAHs (Lohmann and
Lammel, 2004; Shahpoury et al., 2016) and NPAHs (Tomaz et al., 2016) is mostly determined
by absorption in POM and adsorption to soot. When comparing polluted air at the continental
site and background with urban influence at the marine site, a strong shift of $\Sigma_6$4rPAH
towards the particulate phase, $\theta$ ≈0.21 vs. 0.07, respectively, is found, while for $\Sigma_{11}$3-4rNPAH
$\theta$ are similar i.e., ≈0.16 vs. 0.21, respectively. This phase partitioning trend of the 4rPAHs
could be explained by sorption to EC, which is a factor of ≈2 higher, but not by OC (only





≈20% higher). In conclusion, these observations consistently indicate that sorption to soot is
less significant for gas-particle partitioning of NPAHs than for PAHs.
While NPAHs were significantly phase-shifted ($\theta = 0.24$ during day-time but $\theta = 0.58$ during
night-time), this was not the case for 4rPAHs ($\theta = 0.18$ during day-time and $\theta = 0.23$ during
night-time). This is in line with the perception that the temperature sensitivity of phase change
is stronger for the substance class with stronger molecular interactions in the condensed phase,
NPAHs. E.g., the enthalpies of phase change between air and OC of FLT and NFLT are -98
and -75 kJ mol$^{-1}$, respectively (OC represented by DMSO; ACD, 2015).
Good agreement is found for the prediction of NPAH partitioning using the multi-phase (3-
phase) ppLFER with most values predicted within one order of magnitude of the observed
values (Fig. 2; quantification of deviations in S2.3.1). The agreement is better than assuming
absorption (into OM) to be the only relevant process ($K_{oa}$ model; see S2.3.2, Fig. S5). The
same was found when studying gas-particle partitioning of NPAHs in urban air (Tomaz et al.,
2016). This supports the perception that gas-particle partitioning of NPAHs is governed by
various molecular interactions with OM, with its polarity being well represented by DMSO,
better than by octanol. Earlier, it had been found for eight 3-4rNPAHs at urban and rural sites
(Li et al., 2016) that the dual model, assuming adsorption (to soot) and absorption (into OM)
predicts better than single adsorption (to the total aerosol surface i.e., Junge-Pankow) or single
absorption ($K_{oa}$) models do.
The interactions with the aerosol matrix of 9NPHE (continental site) and 5NACE, 2NFLN,
2NFLT and 1NPYR (marine site) are less well represented than other NPAHs by the model as
suggested by low slopes of their log $K_{p\ experimental}$/log $K_{p\ predicted}$ relationships. Further
conclusions are not supported by the limited amount of data and uncertainties on both the
model (estimated ppLFER parameters) and experimental (concentrations close to LOQ) sides.

**Mass size distributions**
Sums of NPAHs' and PAHs' mass size distributions are found unimodal with the maximum
in particles <0.49μm, except PAHs at the marine site, which shows a second maximum
between 1.5 and 3.0 μm (Fig. 3). At the marine site, 50 and 69% of 1NPYR and 2NFLT,
respectively, were found associated with particles <0.45μm and 68 and 86%, respectively,
with particles <0.95 μm, and even more, 83% and 100%, respectively, with particles <0.45μm
at the continental site.



$\Sigma_6$4rPAH mass size distributions are shifted to larger particles in background with urban
influence as compared to marine background air (both collected at the marine site) i.e., MMD
= 0.19 and 0.28, respectively. However, such a trend is not apparent for NPAHs (Table 2). The
size shift of PAHs is not corresponding to the $PM_{10}$ mass size distribution: The MMD of $PM_{10}$
for all samples collected at the marine site was 0.58 µm, while it was 1.13 and 0.62 µm in the
marine background and background with urban influence data subsets, respectively. The $PM_{10}$
as well as the OC mass size distributions were bimodal with maxima corresponding to < 0.49
µm and 3.0-7.2 µm particles (MMDs listed in Table 2), while the EC mass size distribution
was unimodal, with the maximum concentration in the finest fraction. At the continental site,
the $\Sigma_{11}$3-4rNPAH mass size distribution was bimodal with maxima corresponding to < 0.49
µm and 7.2-10 µm particles, while the $\Sigma_6$4rPAH mass size distribution was unimodal, with the
maximum concentration in the finest fraction (for all samples as well as for day and night data
subsets; Table 1).
The formation of a second maximum, at larger particles than emitted, reflects the
redistribution of semivolatile organics in an aged aerosol, hence, is expected at receptor sites
such as the marine site. This was also observed in polluted air, at rural and suburban sites, but
not at traffic sites or in winter at a rural site, when primary emissions dominated (unimodal;
Albinet et al., 2008b; Ringuet et al., 2012b).

**367   Substance patterns and NPAH formation during long-range atmospheric transport**

Among the targeted NPAHs and apart from NNAPs, which were highest concentrated,
2NFLT and 3NPHE prevailed at the marine site (accounting together for ≈60% of the NPAH
mass, excluding the NNAPs), while at the continental site 9NANT and 2NFLT prevailed
(accounting for ≈65% together) (Fig. 1, summarised in Fig. S4). The analytical method did not
separate the isomers 2NFLT and 3NFLT, but at receptor sites, far from diesel emissions it
appears justified to assume $c_{2NFLT} \gg c_{3NFLT}$ (Finlayson-Pitts and Pitts, 2000; Zimmermann et
al., 2012). The ratio 1NPYR/2NPYR is higher, ≈1, at the continental site than at the marine
site (≈0.25), which reflects the significance of primary sources for polluted air (Atkinson and
Arey, 1994; Finlayson-Pitts and Pitts, 2000; Zimmermann et al., 2012). This ratio was found
similarly high or even higher at urban sites (Ringuet et al., 2012c; Tomaz et al., 2016).
Similarly, the ratio 2NFLT/1PYR, the concentration of a secondarily formed over a primary
emitted NPAH, has been used as indicator for fresh emissions (if < 5) vs. photochemically



aged air mass (Keyte et al., 2013). These values were >> 5 in 21 out of 22 and 7 out of 8
samples at the marine and continental sites, respectively. The only sample collected at the
continental site with elevated primary NPAH (2NFLT/1NPYR = 4.3) was possibly influenced
by emissions from Budapest, which was passed by the advected air within the last hours
before arrival. The only sample collected at the marine site with elevated primary NPAH
(2NFLT/1NPYR = 5.9) was indeed directly influenced by emissions into the boundary layer
above the Izmir and Istanbul metropolitan areas (urban fractional dose $D_u$ = 5.0% for samples
no. 1 and 2 in Fig. S3). In conclusion, these results from receptor / background sites confirm
the existing knowledge about primary emitted and secondarily formed NPAHs.
The ratio of two secondarily formed NPAHs, 2NFLT/2NPYR, is indicative for day- vs. night-
time formation paths (Atkinson and Arey, 1994; Ciccioli et al., 1996) is found ≈2 at the
marine and ≈8 at the continental site (normalised to the precursor ratio i.e.,
2NFLT/2NPYR/(FLT/PYR); Table 4). Such low values point to day-time (OH initiated)
formation, while night-time ($NO_3$ initiated) formation was negligible, practically excluded at
the marine site. This is in line with the perception that $NO_3$ must have been very low in this
remote environment. ($NO_x$ levels at the marine site were in the range 0.2-0.6 ppbv). A similar
conclusion had been drawn in a semi-rural environment (Feilberg et al., 2001).
For 2NFLT and 2NPYR (secondary sources only) and for 1NPYR, which has mostly primary
sources (Finlayson-Pitts and Pitts, 2000; Ringuet et al., 2012a; Jariyasopit et al., 2014a,
2014b) we infer the potential yields (Table 4). Here, yield is defined as $c_{NPAH}/c_{PAH}$ (total
concentrations) and reflects an upper estimate, as other PAH photochemical sinks, such as
formation of oxy-PAHs, are neglected. The yield of 2NFLT in polluted air exceeds the one in
background air only slightly, while the yield of 2NPYR in polluted air exceeds the one in
background air much more (a factor of 3 higher).
As expected, the highest potential yield of 1NPYR is found in polluted air (both sites),
reflecting the dominance of primary emissions of 1NPYR. Similarly, higher yields of
secondary NPAHs are found for marine background air compared to background air with
urban influence (marine site), reflecting the longer reaction times elapsed since PAH
emission. The yield for 2NFLT, $c_{2NFLT}/c_{FLT}$, ≈ 2-4% at both sites ranges higher than the one
for 2NPYR, $c_{2NPYR}/c_{PYR}$, which is found ≈ 0.5-2%. Note that because of the co-elution of
2NFLT and 3NFLT, and neglect of 3NFLT, the so derived values of $c_{2NFLT}/c_{FLT}$ represent
actually upper estimates. Apart from sites which were immediately influenced by PAH sources



(road traffic, power plant, biomass burning), only very few studies reported NPAH together
with precursor data in both phases of ambient air. $c_{2NPYR}/c_{PYR} = 1.0\%$, similar to our finding at
remote sites, but a very high $c_{2NFLT}/c_{FLT} = 12.9\%$ were reported from a suburban site in France
in summer during day-time (corresponding values for night-time were 2.0 and 9.4%,
respectively; Ringuet et al., 2012c). 2NFLT was not separated from 3NFLT (similar to our
data set). A suburban site will be influenced by direct 3NFLT emissions, such that $c_{2NFLT}/c_{FLT}$
is an upper estimate. Much lower ratios, $c_{2NFLT}/c_{FLT} = 0.20\%$ and $c_{2NPYR}/c_{PYR} = 0.08\%$ were
reported as the median values for 90 sites of various categories, rural and urban, in northern
China in summer (Lin et al., 2015). These yields are somewhat higher for the subset of the
rural sites. The potential yields found at the marine site in our study are close to the yields for
OH-initiated photochemistry observed in laboratory experiments under high $NO_x$ conditions
i.e., 3% for $c_{2NFLT}/c_{FLT}$ and 0.5% for $c_{2NPYR}/c_{PYR}$ (Atkinson and Arey, 1994).

**4. Conclusions**
For the first time pollution contained in individual background air samples was quantified, by
means of a fractional dose. The fractional dose indicated how much the collected volume of
air had been exposed to an urban boundary layer within a given time horizon. This is found
suitable to discriminate among samples and discuss results, clearly beyond qualitative
reasoning on back trajectories alone. The concept could be applied to any type of
georeferenced origin and might be useful to track the influence of land use of various kind, or
ship and aircraft routes.
Our measurements confirmed occurrence of mutagenic NPAHs, earlier reported from polluted
atmospheric environments of America, Europe and Asia, also for the European background
atmosphere and the outflow of Europe. These substances obviously go into intercontinental
transport and might be indeed ubiquitous. The long-range transport potential is hardly limited
by the mass size distribution, which is determined by the particle size upon emission (primary
NPAHs) and condensation and redistribution in the aerosol along transport, hence, does not
include the short-lived coarse mass fraction. However, the observation of 3.8 and 0.92 pg m$^{-3}$
of 2NFLT and 2NPYR, respectively, measured at the southeastern outflow of Europe (the
lowest ever reported concentrations; Table 3), may indicate that their abundance in the remote
global environment could be less than anticipated. Earlier, this was based on a single
measurement of 2NPYR, 3 pg m$^{-3}$, at an extremely remote site in central Asia two decades



before (Ciccioli et al., 1996). Moreover, this air, classified as marine background, was not
completely clean, but had been exposed to a non-zero fractional urban pollution dose (0.4% of
the total, time horizon of 2 days). More measurements at remote sites should verify levels
globally. PAHs have been abated significantly in Europe during the last decades (EEA, 2014),
which should also be reflected in long-term trends of their derivatives. However, a temporal
trend for the Aegean or the southeastern outflow of Europe in summer cannot be inferred
based on the current and the earlier (2002; Tsapakis and Stephanou, 2007) campaign data.
NPAHs should be included in monitoring programmes to better assess the exposure of human
health hazards of atmospheric pollution, even in remote areas.
Understanding of NPAH formation in ambient air is still rudimentary. Although our
observations of a potential NPAH yield are in agreement with laboratory studies of OH-
initiated photochemistry, the kinetics of NPAHs, both formation from parent PAHs and
photolysis remains to be quantitatively studied under relevant conditions for the background
atmosphere i.e., low $NO_x$ and on various aerosol matrices including sea salt, respectively.

**Acknowledgements**
We thank Christos I. Efstathiou (Masaryk University), András Hoffer, Gyula Kiss (MTA-PE
Air Chemistry Research Group, Veszprém), Jiři Kohoutek (MU), Giorgos Kouvarakis
(University of Crete, Iraklion) and Lajos Szöke (Hungarian Meteorological Service) for on-
site support, Giorgos Kouvarakis and Krisztina Labancz (Hungarian Meteorological Service)
for meteorological and trace gas data, Michael H. Abraham (University College London) for
providing ppLFER solute descriptors, Ignacio Pisso (NILU, Kjeller, Norway) for model post-
processing scripts and Céline Degrendele (MU) and Manolis Tsapakis (Hellenic Centre for
Marine Research, Gournes) for discussion. This research was supported by the Czech Science
Foundation (n° P503 16-11537S), the Czech Ministry of Education, Youth and Sports (n°
LO1214 and LM2015051), and the European Union FP7 (n° 262254, ACTRIS).

**Compliance with Ethical Standards** No potential conflicts of interest (financial or non-
financial) exist.

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



Table 1. Overview time weighted mean concentrations in the particulate and gas phases, and ambient temperature. Data subsets (B = background, P = polluted, D = day mean, N = night mean) and mass size distribution (<0.45/0.45-0.95/0.95-1.5/1.5-3.0/3.0-7.2/7.2-10 μm aerodynamic equivalent diameter) in brackets.

| Site | Phase | $\Sigma_{11}$ 3-4rNPAH (pg m$^{-3}$) | $\Sigma_6$ 4rPAH (pg m$^{-3}$) |
|---|---|---|---|
| Marine | particulate (n = 8) | 4.1 (3.5/0.6/0.2/0.03/0.03/0.00) (B: 0.2/P: 8.7) | 43 (28/8.1/1.2/6.2/4.3/1.7) (B: 7.9/P: 42.4) |
| | gas (n = 21) | 18.4 (B: 13.2/P:31.7) | 403 (B:321/P:595) |
| | T(°C) | 25.6 (B: 27.1/P: 22.0) | |
| Continental | particulate (n = 22) | 24.3 (20.5/2.9/0.7/0.04/0.06/0.15) (D:13.9/N:34.6) | 129 (87/28/12/0.6/0.0/0.0) (D:146/N:116) |
| | gas (n = 22) | 34.2 (D:42.9/N:25.5) | 517 (D:649/N:384) |
| | T(°C) | 23.1 (D:28.8/N:17.5) | |



Table 3. Comparison of total (g + p) concentrations in air, $c_{tot}$ (pg/m$^3$), with other
measurements at remote and rural sites

| | 1NPYR (pg/m$^3$) | 2NFLT (pg/m$^3$) | 2NPYR (pg/m$^3$) | References |
|---|---|---|---|---|
| background CEu summer 2013 | 1.1 | 15 [a] | 1.3 | this work (continental) |
| E Mediterranean summer 2012 | 0.74 | 8.6 [a] | 2.5 | this work (marine) |
| E Mediterranean clean summer 2012 | 0.21 | 3.8 [a] | 0.92 | this work (marine background [b]) |
| E Mediterranean clean summer 2002 | - | 29 | 21 | Tsapakis and Stephanou, 2007 |
| Ross Sea coast, Antarctica | <0.02 [c] | <0.03 [c] | | Vincenti et al., 2001 |
| Himalayas, Nepal 1991 | - | - | 3 | Ciccioli et al., 1996 |
| Forest Amazonia 1993 | 2 | 15 | 8 | |
| Rural Northern Germany 1991 | - | - | 3 | |
| Rural Denmark winter-spring 1982 | 9±5 [c] | - | - | Nielsen et al., 1984 |
| Semi-rural Denmark all year 1998-99 | 40 | 97 | 6.3 | Feilberg et al., 2001 |
| Remote Alps 2002 | 2.2 | - | - | Schauer et al., 2004 |
| Rural Alps 2002 | 6.6 | - | - | |
| Rural Alps [d] winter 2002-03 | 21 | 96 [a] | 81 | Albinet et al., 2008a |
| Rural Alps [d] summer 2003 | 4.2 | 28 [a] | 5.7 | |
| Rural Southern France 2004 | 0.6 | 2.6 [a] | 1.6 | Albinet et al., 2007 |

[a] co-eluted with 3NFLT, assuming $c_{3NFLT} = 0$
[b] samples No. 9, 10, 19 and 22 in Fig. S3
[c] particulate phase concentration only
[d] Val de Maurienne sites (Albinet et al., 2008a)



Table 4: Selected 4-ring PAHs and primary and secondary 3-4 ring NPAH total (g + p) time-
weighted mean concentrations $\pm\sigma$ (pg m$^{-3}$). Potential yields, $c_{NPAH}/c_{PAH}$, in brackets. $\sigma$ given
for n > 2.

| Site | | Marine | | | Continental |
|---|---|---|---|---|---|
| | Data subset | all (n = 8[a]) | marine background (n = 2[b]) | background with urban influence (n = 2[c]) | all (n = 22) |
| Primary | FLT | 213±161 | 161 | 259 | 342±215 |
| | PYR | 146±130 | 103 | 188 | 226±131 |
| Primary and secondary (potential yield) | 2NFLN [d] | 0.038±0.12 | <0.18 | 0.15 | 0.034±0.044 |
| | 1NPYR | 0.62±1.1 (0.4±0.2%) | 0.21 (0.2%) | 1.4 (0.7%) | 1.1±0.6 (0.6±0.3%) |
| Secondary (yield) | 2NFLT [e] | 7.7±8.5 (3.6±2.0%) | 1.68 (1.0%) | 11.0 (4.2%) | 15±10 (6.5±7.5%) |
| | 2NPYR | 2.2±2.6 (1.5±0.7%) | 0.92 (0.9%) | 3.3 (1.8%) | 1.3±1.7 (0.74±1.09%) |

[a] 8 filter and 21 PUF samples
[b] 2 filter and 6 PUF samples i.e., No. 9-10 and 19-22 in Fig. S3 (urban fractional dose $D_u$ =
678  0.4%)
[c] 2 filter and 5 PUF sample i.e., No. 1-2 and 15-18 in Fig. S3 ($D_u$ = 3.1%)
[d] no yield given as $c_{FLN}$ not quantified
[e] co-eluted with 3NFLT, assuming $c_{3NFLT} = 0$



Fig. 1: Time series of absolute (a, c; pg m$^{-3}$) and relative (b, d) total (gas + particulate) NPAH
concentrations at the (a, b) marine (24 h means shown[a]) and (c, d) continental site (day / night
means)
a.                                                                      b.

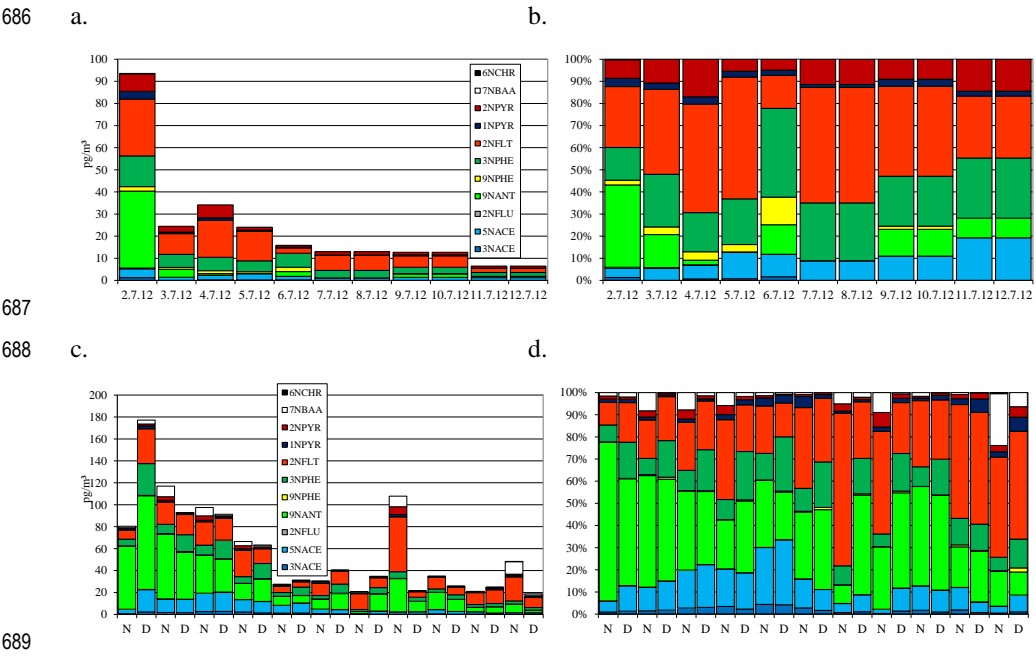


c.                                                                      d.

[a] gas-phase (PUF) sampled day / night, particulate phase (filter) sampled 1-4 subsequent days /
nights, 4 during the period 7.-12.7.12




Fig. 2: Predicted versus experimental log $K_p$ ($m^3$ air $g^{-1}$ PM) for NPAHs using the multi-phase
ppLFER model at the (a) marine and (b) continental site
a.

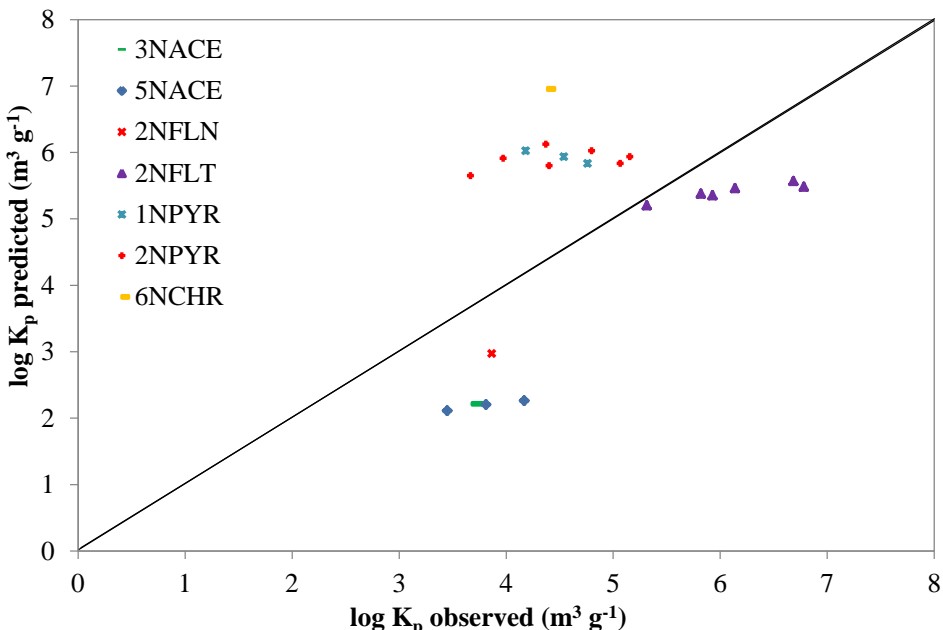

b.



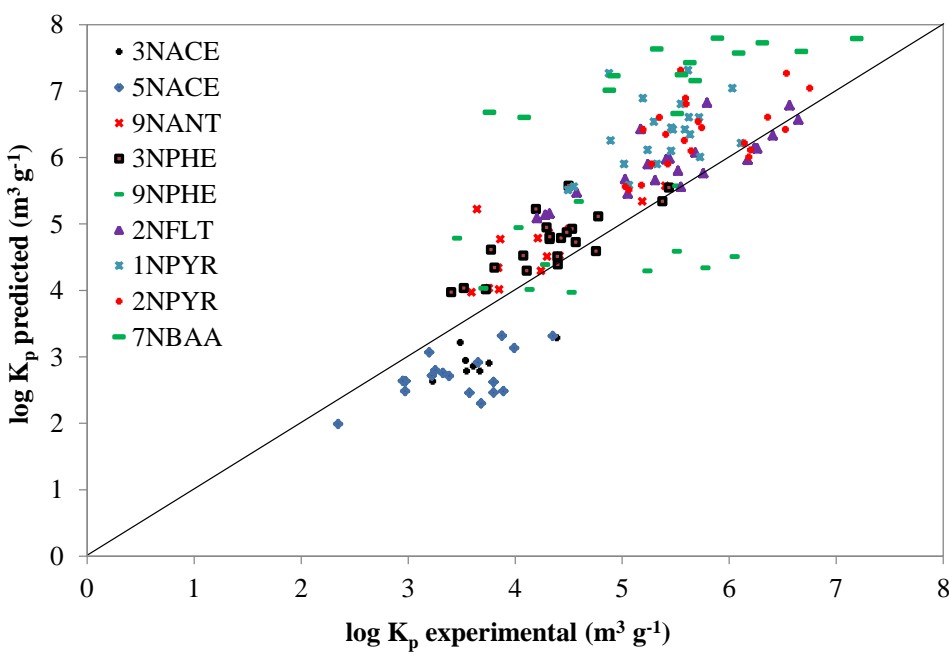






Fig. 3. Time-weighted mean $\Sigma_6$4rPAH and $\Sigma_{11}$3-4rNPAH mass size distributions (%) at the
marine and continental sites. Upper cut-off of impactor stage given in µm of aerodynamic
particle size.

| Site | PAH | NPAH |
|------|-----|------|
| Marine | | |
| Continental | | |





Table 2. Total (g + p) time-weighted concentrations, $c_{tot}$ (pg m$^{-3}$), particulate mass fraction, $\theta = c_p / c_{tot}$, and mass median diameter (MMD, μm), of 2-4-ring NPAHs and 4-ring PAHs at the marine (as 'mean (background mean/ urban influence mean)', n = 8(2$^a$/2$^b$)) and continental (as 'mean (day mean/ night mean)', n = 22(11/11)) sites, together with temperature and supporting aerosol parameters (PM$_{10}$ and carbonaceous mass fractions). LOQ = limit of quantification, nd = no data.

| | Marine | | | Continental | | |
|---|---|---|---|---|---|---|
| | $c_{tot}$ (pg m$^{-3}$) | Θ | MMD (μm) | $c_{tot}$ (pg m$^{-3}$) | Θ | MMD (μm) |
| FLT | 226 (161/259) | 0.07 (0.03/0.08) | 0.58 (0.43/0.52) | 342 (432/251) | 0.11 (0.11/0.12) | 0.062 (0.101/0.034) |
| PYR | 158 (103/188) | 0.04 (0.01/0.06) | 0.21 (0.022/0.22) | 226 (276/176) | 0.18 (0.18/0.19) | 0.075 (0.105/0.055) |
| BBN | 4.1 (2.0/5.5) | 0.01 (nd/0.06) | 0.022 (nd/0.022) | 15 (16/13) | 0.61 (0.58/0.65) | 0.079 (0.127/0.053) |
| BAA | 2.8 (<LOQ/3.4) | 0.28 (nd/0.35) | 0.022 (nd/0.022) | 16 (14/18) | 0.91 (0.90/0.92) | 0.070 (0.090/0.060) |
| TPH | 12 (8.5/14) | 0.02 (nd/0.06) | 0.022 (nd/0.022) | 23 (26/21) | 0.51 (0.41/0.63) | 0.070 (0.090/0.057) |
| CHR | 23 (10/29) | 0.22 (0.09/0.25) | 0.15 (0.022/0.15) | 41 (44/38) | 0.75 (0.71/0.80) | 0.074 (0.105/0.055) |
| Σ$_6$4rPAH | 426 (284/499) | 0.07 (0.02/0.07) | 0.31 (0.19/0.28) | 663 (808/517) | 0.21 (0.19/0.25) | 0.071 (0.10/0.051) |
| 3NACE | 0.21 (0.17/0.39) | 0.05 (0.00/0.19) | 0.022 (nd/0.022) | 1.0(1.0/1.0) | 0.05 (0.01/0.11) | 0.022 (nd/0.022) |
| 5NACE | 1.8 (1.5/2.0) | 0.07 (0.00/0.00) | 0.022 (nd/nd) | 6.8 (7.6/6.0) | 0.03 (0.01/0.05) | 0.022 (0.022/0.022) |
| 2NFLN | 0.01 (<LOQ/0.15) | 0.02 (nd/0.00) | 1.19 (nd/nd) | 0.035 (0.035/0.034) | 0.00 (0.00/0.00) | nd |
| 9NPHE | 0.73 (0.84/0.55) | 0.00 (0.00/0.00) | nd | 0.21 (0.28/0.13) | 0.36 (0.43/0.20) | 0.022 (0.022/nd) |





Atmospheric Chemistry and Physics — Open Access — Discussions — EGU

| | | | | | | |
|---|---|---|---|---|---|---|
| 3NPHE | 4.8 (3.4/5.0) | 0.00 (nd/nd) | nd | 7.4 (10.0/4.8) | 0.24 (0.15/0.44) | 0.109 (0.067/0.116) |
| 9NANT | 4.2 (1.1/8.2) | 0.00 (0.00/0.00) | nd | 22 (22/22) | 0.23 (0.14/0.33) | 0.022 (0.022/0.022) |
| 2NFLT[c] | 8.6 (3.8/11) | 0.32 (nd/0.48) | 0.040 (nd/0.080) | 15 (13/18) | 0.78 (0.54/0.95) | 0.054 (0.035/0.050) |
| 1NPYR | 0.75 (0.21/1.4) | 0.33 (0.00/0.61) | 0.061 (nd/0.14) | 1.1 (1.1/1.2) | 0.82 (0.76/0.88) | 0.030 (0.031/0.029) |
| 2NPYR | 2.5 (0.92/3.3) | 0.53 (0.19/0.70) | 0.058 (0.060/0.055) | 1.3 (0.73/2.0) | 0.93 (0.83/0.97) | 0.070 (0.040/0.061) |
| 7NBAA | <LOQ | nd | nd | 2.5 (0.77/4.2) | 0.91 (0.56/0.97) | 0.074 (0.038/0.057) |
| 6NCHR | 0.02 (<LOQ/0.07) | 1.00 (nd/1.00) | 2.12 (nd/2.12) | 0.01 (<LOQ/0.02) | 1.00 (nd/1.00) | 0.022 (nd/0.022) |
| $\Sigma_{11}$3-4rNPAH | 23.7 (11.8/32.0) | 0.22 (0.01/0.21) | 0.34(0.33/0.34) | 58 (56/59) | 0.16 (0.13/0.17) | 0.039 (0.036/0.040) |
| $PM_{10}$ (µg/m³) | 34.9 (21.0/55.5) | | 0.58 (1.13/0.62) | 22.1 (19.5/24.5) | | nd |
| EC (µg/m³) | 0.11 (0.09/0.17) | | 0.03(0.05/0.03) | 0.31 (0.28/0.33) | | 0.21(0.19/0.22) |
| OC (µg/m³) | 1.9 (1.5/3.0) | | 0.17(0.16/0.15) | 3.6 (3.3/3.9) | | 0.16(0.13/0.18) |
| T (°C) | 25.6 (27.0/22.2) | | | 23.1 (28.8/17.5) | | |

[a] 2 filter and 4 PUF samples i.e., No. 9, 10, 19 and 22 in Fig. S3
[b] 1 filter and 1 PUF sample i.e., No. 1 and 2 in Fig. S3
[c] co-eluted with 3NFLT, assuming $c_{3NFLT} = 0$