# Peer review of "Nitro-polycyclic aromatic hydrocarbons - gas-particle partitioning, mass size distribution, and formation along transport in marine and continental background air 2 3 Gerhard Lammel 1,2\*, Marie D. Mulder 1, Pourya Shahpoury 2, Petr Kukučka 1, Hana Liško"

_Atmospheric Chemistry and Physics, 2016_

## Referee Comment (RC1) · Anonymous Referee #1 · 23 Feb 2017

This manuscript provides a study of nitro-PAHs in 2 background locations. The study includes results about their atmospheric concentrations, gas/particle partitioning measurements and modeling, particle size distribution and substance patterns together with an evaluation of the urban influence of the samples using a Lagragian particle dispersion model. Overall, the manuscript results are scientifically relevant and well presented. The modeling approach used to track the air mass history and to understand the nitro-PAHs sources and processes is very interesting and inovative. Thus, I recommend the publication of this paper in ACP after some minor revisions listed below.

[Figure]

1. Introduction. - Line 46: OK to cite review papers but you should also cite some key papers. - Line 47: mutagenicity yes. Toxicity, no: What about metals, PAHs, other POPs....? - Line 55: recent paper about that: (Keyte et al., 2016) - Line 59-60: OK for 2-NFlt. Instead of 3-NFlt, better to give as an example for primary source, 1-nitropyrene. - Line 68: 2007 instead of 2006 - Line 72: an altitude (remote) site in the French Alps has been also investigated in (Albinet et al., 2008).

2. Methodology 2.2. Chemical analysis. - Line 174: PM10 determined by gravimetry using dedicated filters? Using automatic measurement instrumentation? Precise method and references.

3. Results and discussion - Lines 261-266: These results refer to the particulate phase. What about the gaseous phase? You should discuss both phases or the total. Heterogenous photolysis is true for PAHs too. You could also have an enhanced formation of nitro-PAHs during night-time. Is the pattern the same for all PAHs and all nitro-PAHs or did you notice some differences for specific compounds? The time trend of some compounds should be shown somewhere as an example. - Line 286, Table 3: what about the altitude (remote) site in the Alps? The concentrations seemed very low. - Line 289: ïĄś should be defined before. This refers to the particulate phase, right? - Lines 293-300: this paragraph should be moved in the "Mass size distributions" section. - Line 344, Figure 3: There is a problem with the x-axis, it should be the opposite or the size distribution shown is wrong. PAH and nitro-PAHs are mainly associated to the finest aerosol particles. You should also show the standard deviation of the particle size distribution for each size range. As it is shown on this figure, you should remove these data from the Table 1. - Lines 374-377: To evaluate the influence of primary or secondary sources of nitro-PAHs, it is better to focus on the 2-NFlt/1-NPyr ratio. - Lines 378-388: What about the day/night variations? Did you notice anything specific showing a predominance of day-time or night-time processes at the continental site? This is then useful to support the findings of the following paragraph (Lines 389-396).

References:

Albinet, A., Leoz-Garziandia, E., Budzinski, H., Villenave, E., 2007. Polycyclic aromatic hydrocarbons (PAHs), nitrated PAHs and oxygenated PAHs in ambient air of the Marseilles area (South of France): Concentrations and sources. Sci. Total Environ. 384, 280-292.

Albinet, A., Leoz-Garziandia, E., Budzinski, H., Villenave, E., Jaffrezo, J.L., 2008. Nitrated and oxygenated derivatives of polycyclic aromatic hydrocarbons in the ambient air of two French alpine valleys. Part 1: Concentrations, sources and gas/particle partitioning. Atmos. Environ. 42, 43-54.

Keyte, I.J., Albinet, A., Harrison, R.M., 2016. On-road traffic emissions of polycyclic aromatic hydrocarbons and their oxy- and nitro- derivative compounds measured in road tunnel environments. Sci. Total Environ.

---

## Referee Comment (RC2) · Anonymous Referee #2 · 1 Apr 2017

The Lammel et al. manuscript reports on measurements, partitioning calculations, and back-trajectory modeling of PAHs and nitro-PAHs at two European sites. The sites were chosen to study the transport of PAHs and nitro-PAHs from urban locations. Samples were collected using high-volume samplers, and particles were collected onto quartz fiber filters and gases using polyurethane foam plugs. Samples were extracted using dichloromethane and extracts were analyzed by gas chromatography-mass spectrometry. Partitioning calculations were performed using a recent poly-parameter linear free energy relationship. Particle trajectories were modeling using HYSPLIT and FLEXPART. There are a lot of high quality data and analyses presented in the manuscript,

and the work should prove to be a valuable contribution to the field. There are no concerns with the methodology employed. The primary concern is with the quality of the representation-the technical writing and organization. There are a lot of grammatical errors and undefined symbols and acronyms. It is hard to follow the results as they are presented, particularly with regard to the discussions and conclusions being presented. I recommend significant attention to the quality and clarity of the presentation, and perhaps another round of reviews following revision.

Several specific suggestions and recommendations are provided below regarding grammar/symbols/abbreviations through the abstract and introduction. The intent here is to provide examples of the types of changes that need to be made throughout the manuscript; all such needed changes are not noted beyond the introduction.

Abstract: Why are the fractional doses not reported for the background site? Or is that being used to characterize the fraction at the marine site? This is not clear in the abstract as written. On line 25, it is suggested to replace "were received" with "were calculated".

Line 22-23: For easier reading, recommend to move "with 2-nitrofluranthene . . . most abundant" out of the parenthesis and to the end of the sentence, and then start a new sentence with "While the concentration. . ."

Line 31: NPYR is not defined or otherwise discussed

Line 36-37: Seems out of place. Recommend to remove or better integrate with rest of paragraph.

Line 49-50: Add comma after "more than one third". On line 50, ambient what? Aerosols?

Line 55: Insert "a" between "only" and "few"

Line 60: Suggest to change "indicative for" to "indicative of" or "indicators for"

[Figure]

Line 63: SVOCs is not defined

Line 69: Suggest removing "However ... studied". It has already been established (lines 66-67) that most NPAH observations are in urban areas. Could then use something like..."Though there are a few studies in unpolluted environments: studies were conducted by...". Also suggest to change all the "were" to "have been" (lines 70-75) as measurements may be ongoing or planned.

Line 66-67: Suggest to change "suspect to" to "expect to"

Line 79-80: Suggest to change "lack of data...obvious" to "However, there is limited NPAH data from remote atmospheric environments..."

Line 80: Revise "aim of study to study"

Line 112-113: What was the collection period of for the QFF samples at K-puszta?

Line 116: Ni is undefined

The comments/suggestions below address the clarity of the manuscript and analyses/conclusions presented. There are some technical questions here, but I think that more of the questions/comments arise from organizational structure and imprecise wording.

Conceptually, the urban fractional dose is clear; but how exactly were they calculated? I think much more time needs to be spend on the discussion of the calculated doses, sensitivities, and uncertainties to support the first paragraph of the conclusions section.

Line 209-213: The discussion of the two OM phases is confusing as written. Is one phase low molecular mass/water soluble OM and the other phase high molecular mass/organic soluble OM?

Line 229: What is meant by "the substance patterns are similar"? Diurnal patterns? Monomer ratios? Monomer fractions? (also Line 254)

Line 241: Assume "this" data set refers to the marine data set? Language throughout could be more specific and precise (see also comment above).

Line 256: What is the evidence for the secondary emissions of 4rPAHs? Are the cited studies part of the same campaign? Or is this an inference based on a prior study? Or using measured monomers or ratios using a method in the cited study?

Line 262-267: Is the day/night ratio of PAHs to n-PAHs sufficiently explained by photochemistry? Or may there be measurement-based explanations as well? For example, do the citations referenced support that the photolysis rates of n-PAHs are faster than those of PAHs? If so, this should be explicitly stated. Is the much higher day/night ratio of PAHs surprising given reported oxidation timescales of hours? What exactly is meant by "the same could be reflected" in line 267? Emission rates are governing ratios (based on line 271)? Or that n-PAHs are photochemically degraded faster than precursors (based on line 275)?

Are the lowest reported remote concentrations due to changing urban emissions? Changing photochemistry? Or improvements of in analytical techniques? Some combination? Further discussion on this point is warranted.

Line 289-290: Particle phase distributions? What does the symbol theta represent?

Was the pp-LFER also used for the PAHs? Discussion focused on n-PAHs. How sensitive are the partitioning results to the assumptions in the pp-LFER? What are the uncertainty bounds on the phase predictions? Are they significant enough to influence any conclusions presented regarding phase partitioning? More discussion on the shallow slopes observed for the marine n-PAHs would be valuable. The model does not seem to be capturing any of the observed variance.

Is there a distinction between observed yield and potential yield? Or are these terms being used interchangeably?

Line 436-437: What is meant by "hardly limited by"?

Line 446-447: "Levels" of what?

---

## Author Comment (AC1) · 12 Apr 2017

*Response to reviewers' comments*
Journal: ACP
Title: Nitropolyaromatic hydrocarbons - gas-particle partitioning, mass
size distribution, and formation along transport in marine and continental
background air
Author(s): Gerhard Lammel et al.
MS No.: acp-2016-1145
MS Type: Research article

We would like to thank the reviewers for their thoughtful reading, comments and questions, which considerably helped to improve this manuscript. We have addressed all comments below and have indicated the corresponding modifications in the revised version of the manuscript which is attached to this response (changes highlighted).

**Colour coding below:**
**(1) comments from Referees,**
**(2) author's response,**
**(3) author's changes**

**Anonymous Referee #1**
This manuscript provides a study of nitro-PAHs in 2 background locations. The study includes results about their atmospheric concentrations, gas/particle partitioning measurements and modeling, particle size distribution and substance patterns together with an evaluation of the urban influence of the samples using a Lagragian particle dispersion model. Overall, the manuscript results are scientifically relevant and well presented. The modeling approach used to track the air mass history and to understand the nitro-PAHs sources and processes is very interesting and inovative. Thus, I recommend the publication of this paper in ACP after some minor revisions listed below.

Line 46: OK to cite review papers but you should also cite some key papers.
Early key papers will be added. Sentence will be re-ordered.

Line 47: mutagenicity yes. Toxicity, no: What about metals, PAHs, other POPs. . ..?
True. Will be corrected

Line 55: recent paper about that: (Keyte et al., 2016) -
This paper is cited (line 55). No change will be made.

Line 59-60: OK for 2-NFlt. Instead of 3-NFlt, better to give as an example for primary source, 1-nitropyrene.
Both 3NFLT and 1NPYR are dominated by the primary sources, i.e. diesel and gasoline (Ciccioli et al., 1989; IARC, 1989) vehicles for 3NFLT, and diesel and gasoline vehicles (Ciccioli et al., 1989; IARC, 1989), coal-fired power plants, Al smelters, carbon blacks (IARC, 1989), and coal stoves (Tang et al., 2005) for 1NPYR. But, based on smog-chamber studies; also heterogeneous oxidation by OH radicals was indicated as a (secondary) source for 1NPYR (Jariyasopit et al., 2014b, mentioned in the text, lines 397-398). No change will be made.

References:
Ciccioli P., Cecinato A, Brancaleoni E, Draisci R, Liberti A: Evaluation of nitrated polycyclic aromatic hydrocarbons in anthropogenic emission and air samples: A possible means of detecting reactions of carbonaceous particles in the atmosphere. Aerosol Sci. Technol. 1989, 10, 296-310.

IARC: Diesel and gasoline engine exhausts and some nitroarenes. IARC Monographs on the Evaluation of Carcinogenic Risks to Humans Vol. 46, International Agency for Research on Cancer, Lyon. 1989, URL: http://monographs.iarc.fr/ENG/Monographs/vol46/volume46.pdf.

Tang N, Hattori T, Taga R, Igarashi K, Yang XY, Tamura K, Kakimoto H, Mishukov VF, Toriba A, Kizu R, Hayakawa K: Polycyclic aromatic hydrocarbons and nitropolycyclic aromatic hydrocarbons in urban air particulates and their relationship to emission sources in the Pan–Japan Sea countries. Atmos. Environ. 2005, 39, 5817-5826.

Line 68: 2007 instead of 2006
Will be corrected

Line 72: an altitude (remote) site in the French Alps has been also investigated in (Albinet et al., 2008).
Will be corrected i.e., added to list of studies in remote environments

Line 174: PM10 determined by gravimetry using dedicated filters? Using automatic measurement instrumentation? Precise method and references.
Will be specified

Lines 261-266: These results refer to the particulate phase. What about the gaseous phase? You should discuss both phases or the total. Heterogenous photolysis is true for PAHs too. You could also have an enhanced formation of nitro-PAHs during night-time. Is the pattern the same for all PAHs and all nitro-PAHs or did you notice some differences for specific compounds? The time trend of some compounds should be shown somewhere as an example.
This is a misunderstanding: These results refer to the total, gas + particulate phases.
Will be specified

Line 286, Table 3: what about the altitude (remote) site in the Alps? The concentrations seemed very low.
Thanks. Results at high altitude site in the Alps will be added to Table 3 and discussed.

Line 289: ïA¿s´ should be defined before. This refers to the particulate phase, right? -
Particulate mass fraction θ. The parameter will be introduced

Lines 293-300: this paragraph should be moved in the "Mass size distributions" section.
Thanks, yes, we agree. Will be moved.

Line 344, Figure 3: There is a problem with the x-axis, it should be the opposite or the size distribution shown is wrong. PAH and nitro-PAHs are mainly associated to the finest aerosol particles. You should also show the standard deviation of the particle size distribution for each size range. As it is shown on this figure, you should remove these data from the Table 1.
Thanks, yes, the x-axes were inverted erroneously. Plots will be corrected. Standard deviation of the mean will be shown together with (absolute) mass size distributions.

Lines 374-377: To evaluate the influence of primary or secondary sources of nitro-PAHs, it is better to focus on the 2-NFlt/1-NPyr ratio.
Indeed, this ratio had been used in the literature to this end, and is applied in this study (lines 431-439). However, nowadays, this appears inappropriate, as heterogeneous oxidation by OH radicals was indicated to be a possible secondary source for 1NPYR (Jariyasopit et al., 2014b, mentioned in the text, lines 397-398).

Lines 378-388: What about the day/night variations? Did you notice anything specific showing a predominance of day-time or night-time processes at the continental site? This is then useful to support the findings of the following paragraph (Lines 389-396).

The day/night variation of the ratio 2NFLT/1NPYR, the concentration of a secondarily formed over a primary emitted NPAH, reflects the combination of the diurnal variations of 1NPYR emissions, 2NFLT formation rate, and photochemical sinks of both substances, all unknown in quantitative terms. The day/night variation was 2NFLT/1NPYR $\approx 12/\approx 16$ for day/night, which, therefore, is considered insignificant. No immediate conclusion could be drawn.

References:
Albinet, A., Leoz-Garziandia, E., Budzinski, H., Villenave, E., 2007. Polycyclic aromatic hydrocarbons (PAHs), nitrated PAHs and oxygenated PAHs in ambient air of the Marseilles area (South of France): Concentrations and sources. Sci. Total Environ. 384, 280-292.
Albinet, A., Leoz-Garziandia, E., Budzinski, H., Villenave, E., Jaffrezo, J.L., 2008. Nitrated and oxygenated derivatives of polycyclic aromatic hydrocarbons in the ambient air of two French alpine valleys. Part 1: Concentrations, sources and gas/particle partitioning. Atmos. Environ. 42, 43-54.
Keyte, I.J., Albinet, A., Harrison, R.M., 2016. On-road traffic emissions of polycyclic aromatic hydrocarbons and their oxy- and nitro- derivative compounds measured in road tunnel environments. Sci. Total Environ.

**Anonymous Referee #2**
The Lammel et al. manuscript reports on measurements, partitioning calculations, and back-trajectory modeling of PAHs and nitro-PAHs at two European sites. The sites were chosen to study the transport of PAHs and nitro-PAHs from urban locations. Samples were collected using high-volume samplers, and particles were collected onto quartz fiber filters and gases using polyurethane foam plugs. Samples were extracted using dichloromethane and extracts were analyzed by gas chromatography-mass spectrometry. Partitioning calculations were performed using a recent poly-parameter linear free energy relationship. Particle trajectories were modeling using HYSPLIT and FLEXPART. There are a lot of high quality data and analyses presented in the manuscript, and the work should prove to be a valuable contribution to the field. There are no concerns with the methodology employed. The primary concern is with the quality of the representation-the technical writing and organization. There are a lot of grammatical errors and undefined symbols and acronyms. It is hard to follow the results as they are presented, particularly with regard to the discussions and conclusions being presented.
I recommend significant attention to the quality and clarity of the presentation, and perhaps another round of reviews following revision. Several specific suggestions and recommendations are provided below regarding grammar/symbols/abbreviations through the abstract and introduction. The intent here is to provide examples of the types of changes that need to be made throughout the manuscript; all such needed changes are not noted beyond the introduction.

Abstract: Why are the fractional doses not reported for the background site? Or is that being used to characterize the fraction at the marine site? This is not clear in the abstract as written.

The reason is that advection to the continental background site was influenced by a too homogeneous source distribution in central Europe (during the campaign). This explanation was given in lines 245-247 of the main text (no change).

On line 25, it is suggested to replace "were received" with "were calculated".

Will be corrected

Line 22-23: For easier reading, recommend to move "with 2-nitrofluranthene . . . most abundant" out of the parenthesis and to the end of the sentence, and then start a new sentence with "While the concentration. . ."
Will be corrected

Line 31: NPYR is not defined or otherwise discussed
True. Acronym will be introduced

Line 36-37: Seems out of place. Recommend to remove or better integrate with rest of paragraph.
True. Will be integrated into rest of paragraph.

Line 49-50: Add comma after "more than one third". On line 50, ambient what? Aerosols?
Yes. Will be specified.

Line 55: Insert "a" between "only" and "few"
Will be corrected

Line 60: Suggest to change "indicative for" to "indicative of" or "indicators for"
Yes, thanks. Will be corrected

Line 63: SVOCs is not defined
True. Acronym will be introduced

Line 69: Suggest removing "However . . . studied". It has already been established (lines 66-67) that most NPAH observations are in urban areas. Could then use something like "Though there are a few studies in unpolluted environments: studies were conducted by. . .".
Will be phrased as suggested.

Also suggest to change all the "were" to "have been" (lines 70-75) as measurements may be ongoing or planned.
Will be phrased as suggested.

Line 66-67: Suggest to change "suspect to" to "expect to"
True. Will be phrased as suggested.

Line 79-80: Suggest to change "lack of data. . .obvious" to "However, there is limited NPAH data from remote atmospheric environments. . ."
Will be phrased as suggested.

Line 80: Revise "aim of study to study"
Thanks. '…to study' will be replaced by '…to characterise'

Line 112-113: What was the collection period of for the QFF samples at K-puszta?
The collection period at the continental site will be specified in the sentence (previously in lines 111-113).

Line 116: Ni is undefined
Yes, thanks. Will be specified.

The comments/suggestions below address the clarity of the manuscript and analyses/conclusions presented. There are some technical questions here, but I think that more of the questions/comments arise from organizational structure and imprecise wording.

Conceptually, the urban fractional dose is clear; but how exactly were they calculated? I think much more time needs to be spend on the discussion of the calculated doses, sensitivities, and uncertainties to support the first paragraph of the conclusions section.

The calculation of the fractional dose was explained in the Supplementary material, S2.1.2. This text (S2.1.2) will be shifted to the main text (new Section 2.5), including all equations used for calculation of the dose. Uncertainties will be addressed. The new text will read:

"2.5 Quantification of urban influence on samples

The potential urban influence for individual samples collected at the marine site was based on the fraction of released Lagrangian particles which travelled through an urban boundary layer. A backward run from the sampling site was performed with Lagrangian particles (i.e. air parcels) being released during the entire sampling period. Three urban areas were considered, i.e. Izmir ($\approx$300 km direct distance, 38.2-38.8°N/26.2-27.3°E), Athens ($\approx$300 km, 37.8-38.1°N/23.5-23.8°E) and Istanbul ($\approx$500 km away, 40.8-41.1°N/28.6-29.5°E).

The urban fractional dose, $D_{u\,i}$, an air mass collected in sample i had received for a given simulation period $\Delta t$ can be derived as:

(4)     $D_{u\,i} = \Sigma_t\, N_{blua}(t) \times \Delta t_{Rblua} / (N_{tot}(t) \times \Delta t_i)$

with $N_{blua}(t)$ = number of virtual particles within the urban boundary layer during the specific time step, model output time resolution $\Delta t_{Rblua}$ = 0.5 h, and $N_{tot}(t)$ = number of virtual particles present during the specific time step. Under the given flow conditions in the region, a 2-day time horizon is considered her. Hence, the simulation period is given as:

(5)     $\Delta t_i = \Delta t_{sample} + 48h$

with $\Delta t_{sample}$ being the sampling time. $D_{u\,i}$ takes values between 0 and 1, corresponding to none or all, respectively, of the entire sample air having crossed the urban boundary layer. The $D_u$ time series with allocation to 3 urban areas is shown in the SM, Fig. S3.

The comparison of urban influence in samples of various sample volume, V, requires normalisation to V, a relative dose (equ. (5), with n = total number of samples collected). Values of $D_{ru\,i}$ may exceed 1.

(6)     $D_{ru\,i} = [\Sigma_n V_n/(nV_i)] \times D_{u\,i}$

The urban fractional dose, $D_{u\,i}$, accuracy is limited by the meteorological input data (here 0.125°×0.125° resolution, hourly) and boundary layer depth calculation. In the FLEXPART model, the latter is done according to Vogelezang and Holtslag, 1996."

Line 209-213: The discussion of the two OM phases is confusing as written. Is one phase low molecular mass/water soluble OM and the other phase high molecular mass/organic soluble OM?

This text will be rephrased. The new text will read: "OM is assumed to be constituted of two separate phases, low to mid molecular mass, both organic soluble and water soluble OM. For these, ppLFER equations for dimethyl sulfoxide-air (representing the low molecular mass range) and for polyurethane ether-air (representing the high molecular mass OM) are used, respectively (Shahpoury et al., 2016)."

Line 229: What is meant by "the substance patterns are similar"? Diurnal patterns? Monomer ratios? Monomer fractions? (also Line 254)

Composition of NPAH mixture. Will be specified.

Line 241: Assume "this" data set refers to the marine data set? Language throughout could be more specific and precise (see also comment above).
Yes. Will be specified

Line 256: What is the evidence for the secondary emissions of 4rPAHs? Are the cited studies part of the same campaign? Or is this an inference based on a prior study? Or using measured monomers or ratios using a method in the cited study?
Yes, the cited studies focussed on the atmosphere-surface exchange during the campaigns. This will be clarified to avoid confusion. The revised sentence (lines 256-259) will read: "The investigation of the diffusive air-surface exchange processes during the measurements presented here showed that 4rPAHs were in fact influenced by secondary emissions, namely throughout day and night from the soil at the continental site (by average 16.3 and 9.3 pg m$^{-2}$ h$^{-1}$ for FLT and PYR, respectively; Degrendele et al., 2016) or occasionally from surface seawater at the marine site (during at least 1 day-time interval out of in total 3 of this data subset; Lammel et al., 2016)."

Line 262-267: Is the day/night ratio of PAHs to n-PAHs sufficiently explained by photochemistry? Or may there be measurement-based explanations as well? For example, do the citations referenced support that the photolysis rates of n-PAHs are faster than those of PAHs? If so, this should be explicitly stated. Is the much higher day/night ratio of PAHs surprising given reported oxidation timescales of hours? What exactly is meant by "the same could be reflected" in line 267? Emission rates are governing ratios (based on line 271)? Or that n-PAHs are photochemically degraded faster than precursors (based on line 275)?
Are the lowest reported remote concentrations due to changing urban emissions? Changing photochemistry? Or improvements of in analytical techniques? Some combination? Further discussion on this point is warranted.
Further elaboration on NPAH sources and sinks is impossible, as not enough is known about NPAH atmospheric fate. In fact, this was a major motivation for this study. While the atmospheric lifetimes of 4rPAHs are limited by homogeneous reactions, no such statement is possible for NPAHs. Their lifetimes are eventually limited by heterogeneous photolysis or heterogeneous thermal reactions, but cannot be quantified, as these rate coefficients depend on PM composition and have almost not studied at all (with the exception of wood smoke particles; Fan et al., 1996). Therefore, it is unknown, whether NPAHs are photochemically degraded faster than their precursors. The day/night ratio of PAHs could not be explained by the sinks (stronger during day-time), but is in agreement with the finding of secondary emissions (stronger during day-time) at the site (discussed in lines 262-263).
Also, NPAH field studies are scarce, such that little data exist for comparison (Table 3) and no conclusions can be drawn from existing data with regard to temporal trends.
In order to put day/night ratios into better context, the sentences discussed here (lines 262-271) will be rephrased; "NPAH lifetimes may be limited by heterogeneous photolysis, but available kinetic data are scarce and limited to few aerosol types" (specification in line 265-266). Furthermore, limited knowledge of NPAH atmospheric fate will be highlighted by adding "…but their atmospheric lifetimes are still unknown." To a sentence in the introduction (namely: "These substances have been suggested as tracers for air pollution on the time scales of hours to days" in lines 61-62); The sentence ".More studies into NPAH atmospheric fate, both field observations and kinetic data, are needed in order to assess and quantify spatial and temporal trends, the long-range transport potential and persistence.." (will be added to the conclusions, after line 457).

Line 289-290: Particle phase distributions? What does the symbol theta represent?
Thanks. Will be introduced at first mentioning (line 190)

Was the pp-LFER also used for the PAHs? Discussion focused on n-PAHs.

In this study gas-particle partitioning of parent PAHs was not studied. However, the ppLFER model used had been applied previously on parent PAHs, which showed good prediction power across three urban and rural sites in Europe and the Mediterranean (Shahpoury et al., 2016).

How sensitive are the partitioning results to the assumptions in the pp-LFER? What are the uncertainty bounds on the phase predictions? Are they significant enough to influence any conclusions presented regarding phase partitioning?

More discussion on the shallow slopes observed for the marine n-PAHs would be valuable. The model does not seem to be capturing any of the observed variance.

In lack of experimental data, the ppLFER model for most NPAHs is based on estimated solute-specific Abraham solvation parameters for the various types of molecular interactions. The estimation is based on a group contribution method (ACD, 2015). Experimentally based descriptors used for one NPAH, i.e. 9-nitroanthracene, lead to better predictions (RMSE = 0.33, log $K_p$ units) than the estimated descriptors (RMSE = 0.75). The sensitivity of assumptions on PM phase composition, made in the model had been tested and it was found that the mixing ratios of the WSOSOM and organic polymers' phases, as well as the soot surface area do not contribute significantly to the discrepancy (<< 1, log $K_p$ units).

The predicted variation of $K_p$ is significantly less than the observed for four NPAHs at the marine and one at the continental site (discussed in lines 335-349). The reason is unknown. Apart from PM properties it might be related to sampling or sample handling, although a number of such issues can be excluded, as same sampler, same temperature range, same sampling protocols applied across sites with both satisfactory and deficient agreement between predicted and observed $K_p$.

Satisfactory and deficient agreement between predicted and observed $K_p$ across sites will be discussed accordingly.

Sentences to be added: "While the sensitivity of assumptions regarding PM phase composition, made in the model do not contribute significantly to the deviations (<< 1, log $K_p$ units), a significant part can be attributed to the usage of estimated solute-specific Abraham solvation parameters (taken from ACD, 2015), in lack of experimentally based descriptors. E.g., for an urban site (Tomaz et al., 2016) it was found that experimentally based descriptors used for 9NPAH lead to better predictions than the estimated descriptors i.e., RMSEs differed by 0.43 log units (…) The reason is unknown. Moreover, sampling or sample handling artefacts cannot be excluded, even so same temperature range, sampler and sampling protocols applied across sites with both satisfactory and deficient agreement between predicted and observed $K_p$."

Is there a distinction between observed yield and potential yield? Or are these terms being used interchangeably?

All observed yields are conservatively considered to be potential yields. This will be better specified by replacing "$c_{PAH}$ (total concentrations) and reflects an upper estimate, as other PAH photochemical sinks, such as formation of oxy-PAHs, are neglected" by "…$c_{PAH}$ (total concentrations). This yield is called 'potential' as it reflects an upper estimate , as other PAH photochemical sinks, such as formation of oxy-PAHs, are neglected".

Line 436-437: What is meant by "hardly limited by"?

Will be rephrased: "The mass size distribution is determined by the particle size upon emission (primary NPAHs) and condensation and redistribution in the aerosol along transport, hence, does not include the short-lived coarse mass fraction. This indicates a high long-range transport potential."

Line 446-447: "Levels" of what?

NPAH levels. Will be specified.

Further correction:

In Table 1 (in the line: 'marine gas') and Table 2 (in the column 'marine θ') minor mistakes (which do not impact on the conclusions) were found and will be corrected.

[revised manuscript text omitted]

---

## Author Response (AR2)

*Response to reviewers' comments*
Journal: ACP
Title: Nitropolyaromatic hydrocarbons - gas-particle partitioning, mass
size distribution, and formation along transport in marine and continental
background air
Author(s): Gerhard Lammel et al.
MS No.: acp-2016-1145
MS Type: Research article

We would like to sincerely thank the editor for the suggestions made to improve the grammar and style

**Editor**

Non-public comments to the Author:
One reviewer has seriously challenged your presentation style. Yet, I can not find a severe deficiency in the presented English that would warrant a rejection or major revision. Likely an accumulated amount of small errors have prompted the reviewer to be so negative. If you do not have access to a native English speaker for grammatical editing, I suggest to include the following improvments for a revision, in addition to the already discussed improvements in your response to the reviewer comments.

Suggested improvements:

Line 87: 2 - 13 July 2012: change to: July 2-13 2012
Line 89: 5-16 August 2013 – similar as above
Line 96 had been demonstrated -> was demonstrated
Line 97 are covered -> are measured?
Line 107: had been cleaned - > were cleaned
Line 129: A glass column….
Line 272: The distance to…
Line 433: confirmed the occurrence…
Line 435: (perhaps a better phrasing): These substances are obviously subject to intercontinental transport and might indeed be distributed ubiquitously.
Line 453: The understanding of….or Our understanding of…
Yes, thanks. All these changes followed.

Line 453: this last and final sentence (Although our observations….) is very convoluted and unclear – I suggest either revision by a native English speaker or to simply rephrase this sentence.

[revised manuscript text omitted]